# Time Trends in Treatment Strategies and Survival of Older versus Younger Patients with Synchronous Metastasised Melanoma—A Population-Based Study in the Netherlands Cancer Registry

**DOI:** 10.3390/cancers14194904

**Published:** 2022-10-07

**Authors:** Daisy van der Ziel, Marloes G. M. Derks, Ellen Kapiteijn, Esther Bastiaannet, Marieke Louwman, Frederiek van den Bos, Simon P. Mooijaart, Johanneke E. A. Portielje, Nienke A. de Glas

**Affiliations:** 1Department of Medical Oncology, Leiden University Medical Center, Albinusdreef 2, 2333 ZA Leiden, The Netherlands; 2Department of Research and Development, Netherlands Comprehensive Cancer Center, Godebaldkwartier 419, 3511 DT Utrecht, The Netherlands; 3Department of Gerontology & Geriatrics, Leiden University Medical Center, 2333 ZA Leiden, The Netherlands

**Keywords:** melanoma, older adults, geriatric oncology, immunotherapy, survival

## Abstract

**Simple Summary:**

Immunotherapy has strongly improved outcomes of patients with metastatic melanoma in recent years, but previous studies have shown that survival of older patients often lacks behind. In this study, we investigated treatment prescription of immunotherapy over time in relation to age and survival. We showed that overall survival has improved in patients with synchronous metastasised melanoma aged <75 years, but not in patients aged 75 years or older. This might be explained by lower prescription rates of immunotherapy in this age group.

**Abstract:**

Around 45% of patients with melanoma are older than 65 years. In recent years, immunotherapy has proven very effective for metastasised melanoma. The aim of this study was to investigate the time trends in treatment strategies and survival in older versus younger patients with synchronous metastasised melanoma. We included all patients diagnosed between 2000 and 2019 from the Netherlands cancer registry. We analysed changes in first-line systemic treatment using multivariable logistic regression models, stratified by age (<65, 65–75, and ≥75). Changes in overall survival were studied using multivariable Cox regression analysis. A total of 2967 patients were included. Immunotherapy prescription increased significantly over time for all age groups (<65 years: 11.8% to 64.9%, *p* < 0.001; 65–75 years: 0% to 68.6%, *p* < 0.001; >75 years: 0% to 39.5%, *p* < 0.001). In multivariable analyses, overall survival improved for patients aged <65 and 65–75 (HR 0.96, 95% CI 0.92–1.00 and HR 0.95, 95% CI 0.89–1.00, respectively), but not in patients over 75 (HR 0.98, 95% CI 0.91–1.05). In conclusion, overall survival has improved in patients with synchronous metastasised melanoma aged <75 years, but not in patients aged 75 years or older. This might be explained by lower prescription rates of immunotherapy in this age group.

## 1. Introduction

In recent years, the incidence of melanoma has strongly increased in the Netherlands [1]. Due to the ageing of Western populations, the number of older adults with melanoma will continue to rise. Currently, around 45% of patients with melanoma are aged 65 years or older at diagnosis. Older patients comprise a heterogeneous group due to differences in concomitant diseases and ageing-related factors such as reduced physical functioning, cognitive status and social support system [2,3,4]. These factors may decrease treatment tolerability, thereby complicating oncological treatments [5].

The survival of patients with metastasised melanoma has dramatically improved in the past decade thanks to new treatments that have become available, including targeted treatments (especially BRAF/MEK inhibitors) and immunotherapy via checkpoint inhibition [6,7]. A recent Young SIOG review showed that trials that investigated these treatments did not find any differences in efficacy nor toxicity outcomes in older patients compared to younger patients [8]. However, there were relatively few older patients included in these trials, and the trial populations were generally highly selected with few comorbidities and a good performance status [8,9,10]. Therefore, it remains unclear if these outcomes can be extrapolated to the general older population, with frequent disabilities or concomitant diseases that make them less fit than the populations that were included in the pivotal trials.

The aim of this nationwide registry study was to assess how treatment strategies of patients with synchronous metastasised melanoma have changed in the past two decades in older compared to younger patients, and to evaluate changes over time in overall survival for age groups, using population-based data from the Netherlands Cancer Registry.

## 2. Materials and Methods

For this study, we used data from the Netherlands Cancer Registry. This registry includes data of all patients in the Netherlands who are diagnosed with cancer, through notification of the national pathology database (PALGA). The national hospital discharge databank, which receives discharge diagnoses of admitted patients from all Netherlands hospitals, completes case ascertainment. Survival status is retrieved by linkage with the municipal population registries and was complete up to December 2019. The study was approved by the review board of the Netherlands Cancer Registry.

Importantly, the Netherlands Cancer Registry only includes information on the primary tumour presentation, which means that patients with metastases after a primary localised melanoma are not registered. Therefore, in the current study, we included patients with synchronous metastasised melanoma.

All patients who were diagnosed with synchronous metastasised melanoma between 2000 and 2019 were included (with any N and T stage). Patients were divided into three age groups (<65, 65–75 and ≥75 years) in order to provide information on the different groups of older adults compared to younger patients. This classification based on age was used in several previous publications as well [11,12,13]. Baseline variables that were available included gender, primary tumour localisation (if known), the number of metastatic sites at presentation, the localisation of metastases, the BRAF mutation of the tumour (registered from 2018 onwards), the number of positive lymph nodes, and the Breslow Thickness of the primary tumour.

The treatment information in the Netherlands Cancer Registry was categorised into receiving treatment registered as a yes/no variable, meaning that there is no detailed information on the types of treatment available for this study. In addition, only the first-line treatment is registered. The following treatments were registered: surgical treatment, radiotherapy, chemotherapy, immunotherapy, and targeted therapy.

### Statistical Analyses

First, we used descriptive statistics (using Chi-Square tests) to describe differences in the baseline characteristics between the three age groups. For variables that included some small subgroups (including primary tumour, the number of metastases, and the number of positive lymph nodes), we performed Fisher exact tests instead of Chi-Square as this is more reliable for these analyses. Second, we depicted changes in the (first-line) treatments per age group over time in graphs and assessed these changes statistically using logistic regression models with the treatment of interest as the outcome, and the year of inclusion as the independent variable. Next, we calculated the overall survival per age group using life tables calculated by the Kaplan–Meier Method. Overall, survival was censored at 1 year in order to account for possible bias that could be the cause of a shorter follow-up in most recent years. Overall survival was depicted in Figure 2. We tested the changes in overall survival over time using Cox Regression Models. Analyses were additionally adjusted for baseline tumour characteristics, including sex, the number of metastases, the number of positive lymph nodes, and the Breslow thickness.

Finally, as sensitivity analyses, we tested whether there was an age-specific effect by adding gender as a covariate in the above-mentioned overall survival analyses.

## 3. Results

### 3.1. Patient Characteristics

Overall, 2967 patients were included. Baseline characteristics are presented in Table 1. The majority of patients were male, especially patients aged 65–74 (60.1% of patients aged <65, 66.0% of patients aged 65–74 and 57.4% of patients aged 75 years or older, *p* < 0.001). In the majority of cases, the location of the primary tumour was not specified (70.7% of patients <65, 68.0% of patients aged 65–74 and 64.8% of patients aged 75 years or older, *p* < 0.001).

### 3.2. Changes in Treatments over Time

Changes in treatment strategies that were given in the first line are depicted in Figure 1 and Appendix A.

Immunotherapy was increasingly prescribed in all age groups, but patients aged ≥75 received immunotherapy as first-line treatment less frequently than the two younger age groups (patients <65: increase from 11.8% in 2000 to 64.9% in 2019, 65–74: increase from 0% in 2000 to 68.6% in 2019 and ≥75: increase of 0% in 2000 to 39.5% in 2019, *p* < 0.001 in all age-groups). Similarly, targeted therapy was increasingly prescribed as first-line treatment in all age-groups, but the largest increase was observed in the youngest patients (patients <65: increase from 0% in 2000 to 34% in 2019, *p* < 0.001, 65–74 years: change from 0% in 2000 to 25.4% in 2018 and 7.1% in 2019, *p* < 0.001, and patients ≥75 increase from 0% in 2000 to 11.8% in 2019, *p* < 0.001).

Chemotherapy prescription as first-line treatment strongly decreased in all age-groups (patients <65: decrease from 29.8% in 2001 to 0% in 2019, *p* < 0.001, 65–74: decrease from 11.5% in 2001 to 0% in 2019, *p* < 0.001 and ≥75: decrease from 13.6% in 2001 to 0% in 2018 and 1.3% in 2019, *p* = 0.028).

### 3.3. Survival Outcomes per Age-Group

Overall survival improved significantly over time in patients <65 years (HR 0.97, 95% C.I. 0.97–0.98, *p* < 0.001 unadjusted, and HR 0.96, 95% 0.92–1.00, *p* = 0.029, Table 2 and Figure 2). In patients aged 65–74, survival did improve (HR 0.97, 95% C.I. 0.96–0.98, *p* < 0.001 unadjusted, and HR 0.95, 95% 0.89–1.00, *p* = 0.046). In patients aged 75 years or older, overall survival did not improve during the evaluated time period (HR 0.99, 95% C.I. 0.98–1.00, *p* = 0.073 unadjusted, and HR 0.98, 95% C.I. 0.91–1.05, *p* = 0.556).

There was no age-specific effect on overall survival (HR 1.19, 95% C.I. 0.90–1.57, *p* = 0.219 for female versus male patients).

## 4. Discussion

This study showed that the survival of older patients (≥75) with synchronous metastasised melanoma has not improved between 2000 and 2019 in contrast to younger patients and an increased prescription of targeted treatments and immunotherapy.

These findings may be explained by several reasons. First, our data show that a smaller proportion of older patients receive immunotherapy or targeted therapy as first-line treatment compared to younger patients. This may suggest the undertreatment of older adults with metastasised melanoma. However, we did not have data on treatments that were given after the first line, meaning that we were not able to assess differences in treatment strategies in more detail.

Second, it is possible that ageing-related deficits (such as poor physical function, malnutrition, comorbidity, and cognitive impairments) may have a stronger effect on survival outcomes than melanoma. It is well-known that the risk of dying from other causes than cancer increases with age [14], although the proportion of so-called competing mortality is not very high in patients with metastasised melanoma with a high cancer-related mortality rate.

Another explanation for the differences in survival gains between age groups might be that ageing-related deficits may increase the risk of treatment toxicity, which may have resulted in early treatment withdrawal in the frailest patients. Again, we were not able to test these hypotheses in our data. Several previous observational studies showed that the incidence of severe adverse events (grade III or higher) seems to be comparable to younger patients [15,16,17], but a recent French study did show that older patients had more grade II adverse events, which led more often to treatment discontinuations [18]. However, in contrast with chemotherapy, it is possible that patients who discontinue treatment early still derive a similar response, as immunotherapy may have an ongoing “on-effect” of the immune system in which its efficacy is not directly related to dosage.

Furthermore, our patient population was different than in previous studies, as we were only able to include patients with synchronous metastasised melanoma. According to the Dutch Melanoma Treatment Registry (DMTR) in which all patients with melanoma who were treated in melanoma treatment centres were included, 28.5% of all patients with metastasised melanoma present with synchronous metastasised disease. Based on data from the DMTR, patients with synchronous metastases do not have a different prognosis compared to patients with metastases during follow-up [11]. Additionally, we would still expect that immunotherapy and BRAF/MEK inhibition should have improved outcomes for these patients as well, which we did not observe in our data.

So far, there is no evidence for a different chance of treatment response in older patients compared to younger patients. Some data even suggest that older patients may have a larger chance of responding to immunotherapy with checkpoint inhibitors, but this has yet to be confirmed [19]. Therefore, we believe that the combination of ageing-related factors and possible undertreatment may have resulted in the lack of survival improvement in the oldest age groups.

Interestingly, our study showed that a high percentage of patients received surgical treatment, although this strongly decreased in all age groups. This was most likely explained by a group of patients who were initially locally treated for a skin melanoma, who had positive sentinel lymph nodes and received additional staging postoperatively in which distant metastases were identified (hence defined as synchronous metastasised disease despite the initial “curative” intent of the treatment). In addition, around 30% received radiotherapy, most likely with palliative intent.

These data are, to our knowledge, the first to compare changes over time in the survival outcomes of younger and older patients with synchronous metastasised melanoma [20]. Compared with previous pivotal immunotherapy trials [20], our data show that the benefits of immunotherapy in a real-world population were limited. These trials obviously only included patients who were selected for treatment and who were “fit” enough to travel to specialised melanoma treatment centres, while our real-life data show that for the general (or not selected) older population, survival did not improve. Furthermore, patients who were included in these trials all had an ECOG performance status of 0 or 1, meaning that they were likely much “fitter” than patients in real life, which may explain the poorer outcome of our study population [8].

In order to further investigate the outcomes of older adults with metastasised melanoma, it is important to study predictors of outcome using geriatric assessment data. We are currently establishing a prospective cohort of older adults with metastasised melanoma who are treated with immunotherapy, in which all patients undergo a geriatric assessment at baseline, and in which we will not only study disease-related endpoints but also investigate the quality of life and physical functioning after diagnosis.

The main strength of our study is the fact that the data were derived from the Netherlands Cancer Registry, meaning that all consecutive patients with synchronous metastasised melanoma in the Netherlands were included, and the data were well-registered by trained personnel. Our study has some important limitations. First, we were only able to include patients with synchronous metastasised disease. Second, treatment data were limited to yes/no variables, and only first-line treatments were available in the registry. However, despite the fact that we are not able to show detailed changes in all lines of treatment, the lack of improvement in survival still indicates that the oldest patients have not benefited from therapy improvements in general. Additionally, there was quite a large proportion of missing data on some of the baseline characteristics (such as Breslow thickness or BRAF status), partly explained because the primary tumour was unknown, and BRAF status was registered only after 2017. Finally, we did not have information on comorbidity or geriatric assessment parameters, nor was the cause of death available.

## 5. Conclusions

In conclusion, overall survival has improved in patients with synchronous metastasised melanoma aged <65 years and 65–75 years, but not in patients aged 75 years or older. This might be explained by factors associated with ageing such as frailty or by lower prescription rates of immunotherapy and targeted therapy in this age group. More research is needed to find suitable treatment strategies and tools for optimal treatment selection for the oldest patients.

## Figures and Tables

**Figure 1 cancers-14-04904-f001:**
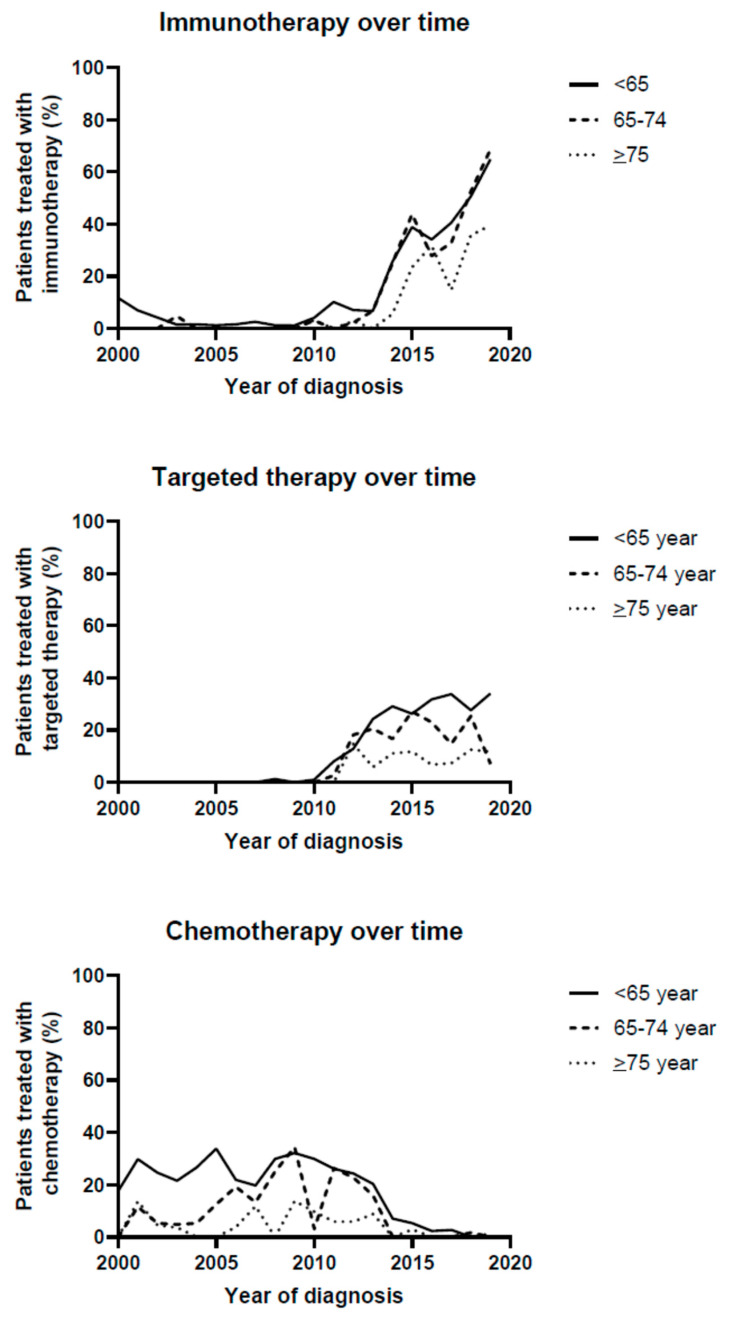
Treatment strategies over time.

**Figure 2 cancers-14-04904-f002:**
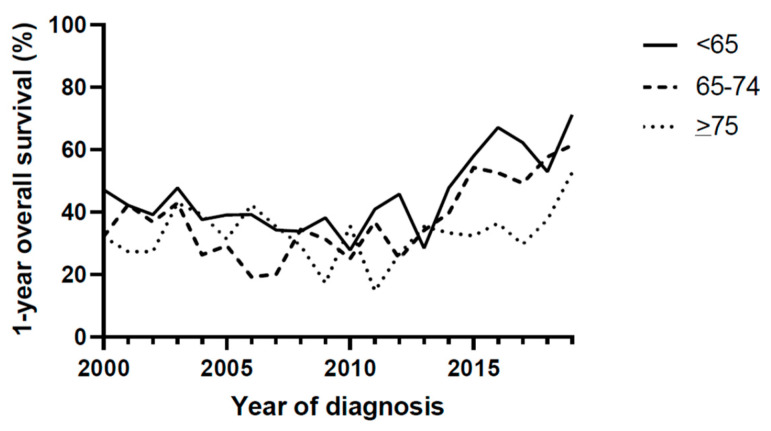
1-year overall survival.

**Table 1 cancers-14-04904-t001:** Patient, tumour, and treatment characteristics by age (<65 vs. 65–74 vs. ≥75).

Patient Characteristics							
	<65		65–74		≥75		*p*-value
	n	(%)	n	(%)	n	(%)	
Gender							
Male	871	(60.2)	458	(66.0)	330	(56.7)	0.002
Female	575	(39.8)	236	(34.0)	252	(43.3)	
Year of inclusion							
2000–2004	307	(21.2)	113	(16.3)	120	(20.6)	<0.001
2005–2009	475	(32.8)	177	(25.5)	143	(24.6)	
2010–2014	319	(22.1)	173	(24.9)	138	(23.7)	
2015–2018	345	(23.9)	231	(33.3)	181	(31.1)	
Primary tumour							
Skin: head	56	(3.9)	36	(5.2)	53	(9.1)	<0.001
Skin: trunk	190	(13.1)	95	(13.7)	60	(10.3)	
Skin: extremities	150	(10.4)	80	(11.5)	86	(14.8)	
Skin: overlay	8	(0.6)	5	(0.7)	0	0	
Skin: not specified	9	(0.6)	2	(0.3)	3	(0.5)	
Female genitals	7	(0.5)	7	(1.0)	7	(1.2)	
Male genitals	0	0	0	0	4	(0.7)	
Unknown	1026	(71.0)	469	(67.6)	369	(63.4)	
Number of metastases							
1	422	(29.2)	184	(26.5)	212	(36.4)	<0.001
2	297	(20.5)	176	(25.4)	137	(23.5)	
3	386	(26.7)	179	(25.8)	104	(17.9)	
4	52	(3.6)	38	(5.5)	21	(3.6)	
5	42	(2.9)	18	(2.6)	9	(1.5)	
6	29	(2.0)	8	(1.2)	6	(1.0)	
7	5	(0.3)	11	(1.6)	6	(1.0)	
8	6	(0.4)	4	(0.6)	0	0	
9	2	(0.1)	2	(0.3)	0	0	
10	1	(0.1)	0	0	0	0	
11	0	0	1	(0.1)	0	0	
Unknown	204	(14.1)	73	(10.5)	87	(14.9)	
BRAF mutation *							
Yes	51	(3.5)	22	(3.2)	14	(2.4)	<0.001
No	21	(1.5)	31	(4.5)	26	(4.5)	
Unknown	1374	(95.0)	641	(92.4)	542	(93.1)	
Positive lymph nodes **							
0	126	(8.7)	69	(9.9)	56	(9.6)	0.314
1 to 5	166	(11.5)	94	(13.5)	65	(11.2)	
6 to 10	14	(1.0)	7	(1.0)	4	(0.7)	
11 to 25	10	(0.7)	7	(1.0)	0	0	
>25	12	(0.8)	2	(0.3)	4	(0.7)	
Unknown	1118	(77.3)	515	(74.2)	453	(77.8)	
Breslow thickness **							
<1	38	(2.6)	22	(3.2)	18	(3.1)	0.001
1–2	49	(3.4)	24	(3.5)	10	(1.7)	
2–4	61	(4.2)	28	(4.0)	16	(2.7)	
>4	116	(8.0)	84	(12.1)	83	(14.3)	
Unknown	1182	(81.7)	536	(77.2)	455	(78.2)	
Immunotherapy							
Yes	209	(14.5)	110	(15.9)	51	(8.8)	<0.001
No	1237	(85.5)	584	(84.1)	531	(91.2)	
Targeted therapy							
Yes	166	(11.5)	76	(11.0)	28	(4.8)	<0.001
No	1280	(88.5)	618	(89.0)	554	(95.2)	
Chemotherapy							
Yes	278	(19.2)	66	(9.5)	23	(4.0)	<0.001
No	1168	(80.8)	628	(90.5)	559	(96.0)	
Surgery							
Yes	732	(50.6)	319	(46.0)	252	(43.3)	0.006
No	714	(49.4)	375	(54.0)	330	(56.7)	
Radiotherapy							
Yes	456	(31.5)	182	(26.2)	139	(23.9)	0.001
No	990	(68.5)	512	(73.8)	443	(76.1)	

* BRAF mutation in patients diagnosed in 2018/2019. ** Patients with a known primary tumour.

**Table 2 cancers-14-04904-t002:** Overall survival by age (<65 vs. 65–74 vs. ≥75). Adjusted analyses were adjusted for gender, tumour location, number of metastases, number of positive lymph nodes, and Breslow thickness.

		HR	95% CI	*p*-value
<65				
	year of inclusion	0.98	0.97–0.99	<0.001
	adjusted for baseline characteristics	0.95	0.91–0.99	0.015
65–74				
	year of inclusion	0.98	0.96–0.99	0.003
	adjusted for baseline characteristics	0.96	0.90–1.01	0.112
>75				
	year of inclusion	1.00	0.99–1.02	0.876
	adjusted for baseline characteristics	0.96	0.89–1.04	0.301

## Data Availability

Data were provided by the Netherlands Cancer Registry: www.cijfersoverkanker.n (accessed on 22 December 2021).

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
