# Peer review of "Time Trends in Treatment Strategies and Survival of Older versus Younger Patients with Synchronous Metastasised Melanoma—A Population-Based Study in the Netherlands Cancer Registry"

_cancers, 2022, doi:10.3390/cancers14194904_

Round 1

Reviewer 1 Report

The aim of this study was to  investigate the time trends in treatment strategies and survival in older versus younger patients with synchronous metastasized melanoma.

This is highly relevant and based on an exhaustive Dutch  registry  of patients with metastatic melanoma.

I congratulates the authors for the article easy to read very clear and well written. 

I have some minor remarks:

-I agree with the authors the boudaries of age groupe but as the cut-off of ageing is not consensual, it may be useful to justify their choices

-Authors said that comparisons were done usong chi-square, but 

In the Table 1, somme numbers are very few, is the Chisqaure appropriate or exact Fisher test may be more appropriate for some comparisons (e.g primary tumors or number of metastases). Alternatively, you can consider to group some categories

- Authors said "patients <65: increase from 11.8% in 2000 to 64.9% in 2019, 65-74: increase from 106 0% in 2000 to 68.6% in 2019 and ≥75: increase of 0% in 2000 to 39.5% in 2019, p<0.001 in all age-groups". I wonder if it may be  add information by adding two-by two  comparison between the 3 age-groups (with alpha correction) to detect if the increase is different between the age groups and not only globally.  It may add some details regarding the difference between age groups

- In the discussion, if I agree that treatement discontinuation  can occur more frequently in older patients,   I do not understand why it can explain the proportion of treated patients as even if it occurs very early, you will count the patient as "treated"  in the proportions 

- May it be interesting to illustrate the Figures by the time point where immunotherapy / targeted therapy was avalaible 

- May it be possible to have the numbers of patients treated by immunoT/targeted therapy and chemotherpay at each time point and by age class? 

- Regarding the Cox Model, did the authors verified the assumptions of the model (residuals, proportionalty) as handling year in a log-linear way is a strong assumption 

- Please remove the term "number of" in the legend of the Survival model before gender.

- Please add the methods (Kaplan-Meier) in the Figures of Overall Survival and the numbers (Deaths/at risk) at each time points 

- Did the authors explore if there is a gender effect?

These remarks are more an interest for the paper than critisicms as the work is interesting, with robust data and smart way to present it event if some details in the analysis can be improved. 

Reviewer 2 Report

This is a very good manuscript. It is well written; conclusions are clear-cut and well discussed.

However, there is a major limitation because patients health status is the “black box”, but the authors discuss appropriately this limitation and give information on future evolution of the data registering.

I have just concern with the sentence line 179-182: “These data are to some extend contradicting with previous trial results of the pivotal immunotherapy trials, which suggested that outcomes of older patients were comparable to younger patients [18]”. There is no contradiction: on the one hand older patients in trials have same outcome than younger patients (but are selected : the health status “black box”); on the other hand in the real life older patients seem not to benefit from immunotherapy (the “black box” includes here not only health status but access to highly specialized centers, institutional decision making procedure and certainly doctor’s decision itself).  

Reviewer 3 Report

Dear authors;

I would like to congratulate you for your work. It's a high number of patients, so this gives relevance to the review

However, its retrospective character dismishes the importance of the review

I don't agree that cut-off ages include the range 65-70 in the same group

As is known by geriatricians, aging changes starts at the age of 70

So, I'll rather prefer that you include an age review in the range 65 to 69 and 70 to 74

This implies re-writting your work

But results and conclusions could be more relevant to scientists

Round 2

Reviewer 3 Report

.